# The Improvement on the Performance of DMD Hadamard Transform Near-Infrared Spectrometer by Double Filter Strategy and a New Hadamard Mask

**DOI:** 10.3390/mi10020149

**Published:** 2019-02-23

**Authors:** Zifeng Lu, Jinghang Zhang, Hua Liu, Jialin Xu, Jinhuan Li

**Affiliations:** 1Center for Advanced Optoelectronic Functional Materials Research, and Key Laboratory for UV-Emitting Materials and Technology of Ministry of Education, Northeast Normal University, 5268 Renmin Street, Changchun 130024, China; luzf934@nenu.edu.cn (Z.L.); jinghang1927@163.com (J.Z.); lijh248@nenu.edu.cn (J.L.); 2Demonstration Center for Experimental Physics Education, Northeast Normal University, 5268 Renmin Street, Changchun 130024, China; 3Changchun Institute of Optics, Fine Mechanics and Physics, Chinese Academy of Sciences, Changchun 130033, China; xujialinseu@163.com

**Keywords:** spectrometer, infrared, digital micromirror device (DMD), signal-to-noise ratio (SNR), stray light

## Abstract

In the Hadamard transform (HT) near-infrared (NIR) spectrometer, there are defects that can create a nonuniform distribution of spectral energy, significantly influencing the absorbance of the whole spectrum, generating stray light, and making the signal-to-noise ratio (SNR) of the spectrum inconsistent. To address this issue and improve the performance of the digital micromirror device (DMD) Hadamard transform near-infrared spectrometer, a split waveband scan mode is proposed to mitigate the impact of the stray light, and a new Hadamard mask of variable-width stripes is put forward to improve the SNR of the spectrometer. The results of the simulations and experiments indicate that by the new scan mode and Hadamard mask, the influence of stray light is restrained and reduced. In addition, the SNR of the spectrometer also is increased.

## 1. Introduction

In the 1970s, the Hadamard transform (HT) was proposed and developed into a relatively mature theory [1]. With the emergence of the mechanical encoding mask, the HT was applied to the near-infrared (NIR) spectrometer. The encoding mask is a key device in spectrometers. However, adopting the mechanical mask, the spectrometer exhibits a complex structure, low resolution, and short life. Compared with the traditional instrument, it possesses no advantage. The development of HT spectrometers is restricted by the encoding mask. Later, the digital micromirror device (DMD) was developed and applied to the HT spectrometer as an encoding mask. Because the HT spectrometer based on the DMD has several advantages such as a higher signal-to-noise ratio (SNR), wider spectral range, and low cost [2,3], DMD-based HT spectrometers have attracted significant research attention.

At present, the performance of HT spectrometers has been greatly improved, but they still have defects, such as the grating diffraction of the spectrometer, the two-dimensional grating diffraction of the DMD, and the poor spectral efficiency of the light source; these defects can make the spectral energy distribution uneven. Thus, the influence of stray light on the absorbance of the whole spectrum is varied; the lower the spectral energy is, the greater the influence by stray light is. The low-energy spectral band exhibits a low SNR, nonlinearity, whereas the high-energy spectral band performs well in those aspects. To improve the energy of the entire spectrum, Wang and colleagues proposed a spectrum-folded structure of a HT spectrometer and a special illumination optical device, but the structure of the spectrometer was complex [4,5]. Zhang et al. proposed a new algorithm to realize energy compensation of the spectrum and analyzed the effect of the HT on the noise without considering the noise distribution [6]. Quan et al. analyzed the spectral distortion in the HT spectrometer and presented a correction approach [7]. However, for the analysis of stray light, especially that with a high correlation of energy distribution, their processing effect was unsatisfactory. Xu et al. analyzed the influence of the HT on the noise before and after coding. They also proposed a new encoding mask to correct the anomaly in the spectra caused by optical defects [8]. With the variation in the height of the stripes, their new encoding mask exhibited a low utilization rate of the DMD.

To improve the performance of the DMD HT NIR spectrometer, a new method of the split waveband scanning is proposed in this paper to mitigate the impact of the stray light. It can not only reduce the influence of stray light on the low-energy spectral bands, but also improve the linearity and accuracy of absorbance in the low-energy spectral bands. On the other hand, a new Hadamard mask of variable width-stripe matching with each scanning area is presented to improve SNR. Based on the new scanning method and coding mask, the simulation and experimental results indicate that the stray light is suppressed and the spectral energy distribution is more uniform. The SNR of the spectrum is also improved, especially in the low-energy spectral band the SNR is increased significantly. It is demonstrated that by the proposed approach, the minimum SNR in the low-energy spectral band is improved by a factor of 7.434 greater than that of the traditional HT method.

## 2. Theory of Hadamard Transform (HT) Spectrometer with Digital Micromirror Device (DMD)

A schematic of the spectrometer designed by us is illustrated in Figure 1a. The incident light emitted from the sample pool is dispersed by the grating, and the dispersion spectrum imaged on the DMD plane by the imaging lens is encoded and reflected. Then, the reflected light is focused onto the detector by the converging lens. Finally, the detector signal is decoded and processed by a computer. Because DMD is programmable, multiple scan modes are available to the spectrometer such as the column scan mode, the Hadamard scan mode, and other multiplexed scan mode. To our spectrometer, the major scan mode is Hadamard scan. The coding matrix of the Hadamard spectrometer is an S-matrix determined by quadratic residue method and can be used to describe the patterns to be displayed on the DMD [9]. By this approach, the spectrum can be modified.

Based on our theory, the HT NIR spectrometer with a DMD can be realized. The photograph of the spectrometer is shown in Figure 1b. The parameters of the spectrometer are listed in Table 1. Before the spectrometer is used, the calibration must be performed to get the relationship between the wavelength and the pixel location in the DMD column. Three lasers whose wavelengths are respectively 1550 nm, 1625 nm, and 2210 nm are used to calibrate approximately the spectrometer, and a Xenon Calibration Light Source of Ocean Optics is used to perform the accuracy calibration. The detailed calibration process of the spectrometer can be found in reference [10].

The spectrum of the light source tested by the spectrometer designed by us is presented in Figure 2. The relative spectral energy is high in the range of 1350–2200 nm and low in the range of 2200–2450 nm. The spectral curves tested by some other spectrometers with different designs may have some differences, but the spectral distribution is the same. The energy in the central wavelength band of the spectrum is high and those on the two edges are low.

## 3. The Measurement Result Increase in the Low-Energy Spectral-Band Absorbance 

### 3.1. Impact of Stray Light on the Spectral Band with Different Energies

Stray light is an important measurement parameter of the spectrometers. The existence of the stray light results in the significant measurement errors [11]. Especially to the dispersive spectrometer, it may cause the nonlinear problems of the instrument. Thus, it is critical to suppress the stray light. In this study, the stray light originates mainly from the scattered light and reflected light of the spectrometer. When the detector receives a certain wavelength signal, it always mixes with some stray light, which is not part of the signal. The existence of stray light will reduce the measured absorbance, especially in spectral bands with strong absorption [8].

This type of stray light is very complex. It may lead to a test result with a homogeneous background. The background values can be measured before beginning the spectral test. When the DMD is closed, the signals received by the detector are the background values. By subtracting the background values from the spectral test signals, the influence of a large proportion of stray light can be corrected. Thus, its performance optimization may be less expensive than other spectrum analysis systems. Beyond that, there is still a fraction of stray light remaining in the energy distribution of the spectrum that cannot be corrected by this method. Using the Zemax software, we can build a nonsequence mode of the optical structure of the spectrometer. By this mode, the energy distribution of the slit images can be obtained in different wavelength bands on the surface of the DMD. The irradiance of the slit image at a wavelength of 1600 nm is shown in Figure 3. In this Figure, the energy of the slit image is the highest and there exist other stray lights with a lower energy around the slit image, except for the background light. Because of the secondary reflection of the devices in the spectrometer, there are two energy circles around the slit image, which may be related to the energy distribution of the slit. The secondary reflection mainly comes from the front and rear surfaces of the DMD window. In addition, ghosting can be observed under the slit image, which is formed by the window of the DMD. Because the micromirrors of DMD are easily broken, the optical window is a must to protect them from the surroundings and permit the light of a certain wavelength range to transmit. The material of the DMD window is a kind of optical glass (Corning 7056) whose main constituent is SiO_2_. For visible, near infrared and ultraviolet wavebands, the anti-reflection films covered on the glasses are different. The refractive index of this type of glass is 1.487 at the wavelength of 545 nm, and the cutoff wavelength for transmission is 2.7 µm. Because the transmission of the DMD window in 1700–2500 nm is reduced, the absorption and reflection are strengthened. So, ghosting can’t be avoided in a carefully designed spectrometer because of the characteristics of DMD. The stray lights formed by the secondary reflection have a great influence on the absorbance measurements of the low-energy spectrum [7].

Therefore, we conduct some analyses to determine the influence of stray light. The absorption spectrum of 95% ethanol is respectively measured by our spectrometer and UV-Visible/NIR Spectrophotometer UH4150 produced by Hitachi High Technologies Corporation (Tokyo, Japan), as shown in Figure 4. From Figure 4, we can see some stronger absorption peaks of the C–H, C–H_2_, and O–H bending and harmonic vibration in ethanol always exist in the long-wavelength band. However, some peak values are less accurate than the reference and some weak peaks cannot be observed. In 2200–2450 nm, the three peaks shown by black line B move about 1–1.5 nm in the direction of long wave and two peaks at 2280 and 2297 nm are not observed compared with the blue line A; In 1400–2200 nm, there is also a red shift of 1.3–3 nm at all peaks except at 1693, 1760, and 1937 nm compared with the reference spectrum. It may be caused by the inaccurate calibration of the spectrometer and the lack of the data points collected by our spectrometer. Comparing the measured spectrum with the reference spectrum, it shows some deviations of absorption intensity between the test values and the reference in whole wavelength band, and the difference of absorption intensities in 2200–2450 nm is greater than that in 1400–2200 nm. An important factor is the stray light. We suppose that when testing the sample spectrum, the low-energy waveband might be disturbed by the stray light from the high-energy waveband, and the high energy waveband will be affected by the stray light from the low energy waveband. The influence of the test results cannot be ignored. 

Then, we perform simulation analysis to verify this hypothesis. The light source is absorbed by the sample. The transmission spectrum is received by the detector, and the energy of the spectrum contains two parts. One is from the transmission light ($T_{\lambda }$) and the other is stray light ($S_{\lambda }$). The absorbance of a certain wavelength of the spectrum can be calculated by the following equations:(1)$$A=\mathrm{lg}\left(\frac{I_{\lambda }}{T_{\lambda }+S_{\lambda }}\right),$$
(2)$$\;S_{\lambda }=\sum_{m}k_{m}T_{\lambda m},$$
where $I_{\lambda }$ is the spectral intensity at wavelength *λ* and $S_{\lambda }$ denotes the total stray light intensity of the high-energy wavelength band (or low-energy wavelength band). $T_{\lambda m}$ is the corresponding intensity of the sampling points of the transmission light and *m* is the spectral sampling number. $k_{m}$ is the proportion coefficient between the stray light intensity and spectral intensity at a given wavelength. Because each proportion coefficient $k_{m}$ is different and complex, we perform simulation experiments under the ideal condition, which state that in the same spectral band, the proportion coefficient $k_{m}$ of the stray light intensity of each sampling point is replaced by the same coefficient *k* (*k* = 0.01, 0.001, 0.0001). To simplify the analysis process, the spectrum is divided into two parts: the high-energy wavelength band (1350–2200 nm) and low-energy wavelength band (2200–2450 nm). By this analysis, we suppose that the stray lights produced by the high-energy wavelength band and low-energy wavelength band have different impacts on each other.

First, we analyze the influence of stray light on the low-energy wavelength band. According to the spectrum presented in Figure 2, the range of spectral intensity $\left(I_{\lambda }\right)$ is between 10,000 and 40,000. Assume that the absorbance value of low-energy wavelength band is fixed, and set it to be 1. Then the intensity of transmission light at a given wavelength: $T_{\lambda }=0.1I_{\lambda }$. In the high-energy wavelength band, the range of $T_{\lambda }$ is between 40,000 and 80,000. By solving Equations (1) and (2), the curve of the intensity effect on the absorbance obtained by the simulation is shown in Figure 5. The purple line is the real absorbance. We can see that the lower the transmission light intensity is, the greater the influence of stray light from the high-energy wavelength band is. The tested values deviate from the real value. Next, we analyse the influence of stray light on the high-energy wavelength band. The range of spectral intensities ($I_{\lambda }$) is between 40,000 and 80,000. Assume that the absorbance value of high-energy wavelength band is fixed, and set it to be 0.155. Then the intensity of transmission light at a given wavelength: $T_{\lambda }=0.7I_{\lambda }$. In the low-energy wavelength band, the range of $T_{\lambda }$ is between 10,000 and 40,000. The curve of the intensity that affects the absorbance obtained by the simulation is shown in Figure 6. From the magnified plot in the inset, the stray light has a small influence on the high-energy wavelength band. Then, we must address the problem that the low-energy wavelength band with the strong absorption is influenced more easily by the stray light than the high-energy wavelength band.

The simulation indicates that the stray light has a great influence on the low-energy wavelength band. In the actual measurement, the experimental conditions are more complicated. The intensity of stray light produced by each sampling point has a different contribution to the transmission light intensity.

### 3.2. Suppression of Stray Light by the Split Waveband Scan Method

According to the analysis results, we propose a split waveband scan method. It is similar to the method of decreasing the stray light radiation. We select a filter and place it between the light source and slit. When we choose the high-pass filter, the signals only appear at the short waveband, and at the long waveband, if some signals appear at the same time, it should be the stray light produced by the short waveband. Whereas, if we select the low-pass filter, the result will be reversed. Thus, we can obtain the entire spectrum by combining the parts of two test results which have no stray light. This method can realize the suppression of the stray light. The switching of the filter is realized in front of DMD window by the rotor controlled by the electric system.

The absorbance curve of the 95% ethanol solution is shown in Figure 7. As the requirements, our team make two types of filters. One is short-wave pass filter whose cutoff wavelength is 2210 nm, and the transmission is higher than 95% in 1350–2200 nm. When the wavelength is greater than 2210 nm, the transmission will be 0.01%. The other is a long-wave pass filter whose cutoff wavelength is 2190 nm, and the transmission is higher than 95% in 2200–2500 nm. When the wavelength is less than 2190 nm, the transmission will be 0.01%.The first filter allows high-relative-power waveband (1350–2200 nm) to pass. The other one allows low relative-power waveband (2200–2450 nm) to pass. When putting the filters into the spectrometer, the scan waveband has been divided into two parts: a high relative-power waveband and a low relative-power waveband. At the high power waveband, there is a noticeable difference between the two curves. This is caused by the decrease of the stray light which originates from the low power band. At the low power waveband, the absorbance value tested by the new method is greater than that of the traditional method. This indicates that the method of the split waveband scan can suppress parts of the impact of the stray light and make the measurement result of the absorbance increase. 

## 4. SNR Improvement of Low Relative-Power Waveband by a New Hadamard Mask

The NIR analysis technique is based on the small change detection in a strong background signal. The level of the SNR will have an impact on the accuracy of the analysis results, which is an important indicator [8]. In part 3, we have already provided a new scan method to suppress parts of the stray light impact, but it can’t improve the SNR of the low-relative-power waveband. Thus, a new Hadamard mask of variable-width stripes is put forward to address the issue.

### 4.1. Impact on SNR of Different Spectral Energies

The noise source of the spectrometer includes the noise from the detector circuit and light source, which determines the SNR of the HT spectrometer [6,12,13,14,15]. Assume that n is the order of Hadamard matrix. The root mean square (RMS) of the illumination noise after the HT will be $\sqrt{\frac{2n}{n+1}}$ times higher than the original [16]. That of the detector signal noise after the HT will be $\frac{2\sqrt{n}}{n+1}$ times higher than the original [15]. Further, the SNR gain becomes $\frac{\sqrt{n}}{2}$ [16]. In [8], Xu et al. concluded that if we want to give priority to select the HT scan mode, some conditions should be satisfied. When the order n is sufficiently large so that the detector noise is equal to the illumination noise, the total noise after the HT will be lower than that after the column scan. Then, the HT scan mode will be correct. 

According to [8], the SNR for the column scan method and S matrix of the HT is respectively expressed: (3)$${\mathrm{SNR}}_{S}=\frac{D}{\sqrt{{\left(D\eta \right)}^{2}+\Delta^{2}}}$$
(4)$${\mathrm{SNR}}_{H}=\frac{D}{\sqrt{{\left(D\eta \times \sqrt{\frac{2n}{n+1}}\right)}^{2}+{\left(\frac{2\sqrt{n}}{n+1}\Delta \right)}^{2}}}$$
where *D* is the real spectral intensity of the light source, *η* is the stability of the light source, Δ is the electronic system noise collected by the analog-to digital converter (AD), and *n* is the matrix order of HT. Based on Equations (3) and (4), we obtain the relationship between relative power intensity and SNR, and the relationship between HT order and SNR, as shown in Figure 8 and Figure 9.

When the order is 107, the RMS of the light source drops off in proportion, and the requirement of the HT scan mode is satisfied. Figure 8 shows the HT scan has an advantage over the column scan method under the weak light intensity. 

When the light source is stable and *n* = 107, the relative power intensity of the spectrum is 10000. The curve of the change in SNR following *n* is shown in Figure 9. When the order *n* increases, the SNR of the two scan methods tends to be decreased. Whereas, compared with the column scan method, the SNR of the HT scan method is decreased slowly. Because the light source energy has a different distribution in the entire waveband, the SNR of the spectrometer will be influenced. We must think of a method to change the condition of the nonuniform distribution of the spectral energy and improve the SNR of the spectral edges with low energy.

### 4.2. Increase in SNR of Low-Energy Waveband by New HT Scan Mask

To overcome the impact, we propose a new coding mask, a Hadamard mask of variable-width stripes (V_W_-Hadamard) in the DMD HT spectrometer. By optimizing the mask, the SNR of the low-energy waveband can be improved. There are three components that determine the resolution: the slit width, the optical transfer function, and DMD resolution. Because of the small pixel size of the DMD, the DMD resolution is not the major determinant [9]. Thus, we plan to change the resolution in different patterns of the DMD. By making sacrifices on the resolution of the edge DMD patterns, the energy loss in the low-energy waveband can be reduced. The width of the scanning stripes is determined by the corresponding spectrum energy in each stripe. The higher the corresponding spectrum energy is, the narrower the stripe width is. Then, the SNR of the edge spectrum with low energy can be improved. Xu et al. proposed a HT scan method by using a mask of variable-height stripes [8]. This method greatly improved the SNR and the even distribution of light source spectral energy. However, because of the change in the height of the stripes, the DMD pattern could not be utilized well. Moreover, a part of the spectral energy was lost. In this study, the V_W_-Hadamard mask can make full use of the spectral energy because of the adaptive variation in the width of the stripes. The new HT mask has an advantage on the energy utilization over the traditional mask. The flow chart of the mask production is outlined in Figure 10.

Assume $E_{m}$ is the median power intensity, and $W_{m}$ is the strip width of *E_m_*, which is taken as the width of three columns of the micromirrors. The width *W* of the other changing strip corresponding to the power intensity of *E* can be
calculated by
(5)$$\frac{E_{m}}{E}=\frac{W}{W_{m}}$$

Whatever the result of the equation is, there exists a maximum wavelength which makes the spectral waveform meaningful. The spectral resolution should not go beyond that. Figure 11a,b separately show that the traditional and the new Hadamard mask. They are produced by the same order of 107.

In order to verify the performance improvement of the HT spectrometer based on the new coding mask, we perform some tests on the light source spectrum. The DMD resolution is 912 × 1104. There are 912 columns of micromirrors in the direction of the spectrum dispersion. The curves of the relative power intensity acquired by the column scan method with different sampling points are presented in Figure 12. With the increase in the number of sampling points, the spectral energy received by the detector decreases. Furthermore, the distribution of each wavelength corresponding to the energy is more even than before. Considering the curves in Figure 9 and Figure 12, the number of sampling points can’t be too large. Thus, we choose the orders of 107 and 227 to perform the test and analysis. The spectrum is divided into two parts: a high-energy wavelength band (1350–2170 nm) and low-energy wavelength band (2170–2450 nm). Based on reference 8, the SNR_vi_ by the new mask scanning method can be calculated by
(6)$${\;\mathrm{SNR}}_{\mathrm{Vi}}=\frac{D_{i}}{\sqrt{{\left(D_{i}\eta \times \sqrt{\frac{2N_{i}}{N_{i}+1}}\right)}^{2}+{\left(\frac{2\sqrt{N_{i}}}{N_{i}+1}\Delta \right)}^{2}}}$$
where *i* denotes the *i*-th scanning stripe, SNR_vi_ is the SNR of the *i*-th stripe region, *D_i_* is the spectral intensity of the *i*-th stripe scanning region, and *N_i_* is the matrix order of the HT in the *i*-th scanning stripe. The curves of the spectral SNR calculated by the equations (Equations (3), (4) and (6)) are shown in Figure 13 and Figure 14. 

For *n* = 107, the average SNR obtained by the V_W_-Hadamard method (${\mathrm{SNR}}_{V}$) in the short-wave sideband is 3289.153, which is 2.034 times that by the scanning spectrometer (${\mathrm{SNR}}_{S}$). That by the traditional Hadamard method (${\mathrm{SNR}}_{H}$) is 3293.121, which is 2.036 times that of ${\mathrm{SNR}}_{S}$. For the average SNR in the long-wave sideband (2220–2450 nm), ${\mathrm{SNR}}_{V}$ is 2.972 times that of ${\mathrm{SNR}}_{S}$ and 1.062 times that of ${\mathrm{SNR}}_{H}$. For the minimum SNR in the short-wave sideband, ${\mathrm{SNR}}_{V}$ is 2.547 times that of ${\mathrm{SNR}}_{S}$ and 1.026 times that of ${\mathrm{SNR}}_{H}$. For the minimum SNR in the long-wave sideband, ${\mathrm{SNR}}_{V}$ is 11.52 times that of ${\mathrm{SNR}}_{S}$ and 2.213 times that of ${\mathrm{SNR}}_{H}$. When *n* = 227, the average SNR of the V_W_-Hadamard spectrometer is improved compared to that of the Hadamard. Particularly, for the minimum SNR in the long-wave sideband, ${\mathrm{SNR}}_{V}$ is 20.397 times that of ${\mathrm{SNR}}_{S}$ and 2.743 times that of ${\mathrm{SNR}}_{H}$. According to the results, the V_W_-Hadamard mask with the order of 227 performs better than that with the order of 107 in the long-wave sideband.

The curves of the DMD resolution by the two methods are presented in Figure 15. Because the DMD resolution matches with the spectral resolution, the spectral resolution (the yellow curve) in Figure 15 is the same as the DMD resolution of the traditional Hadamard. From Figure 15, we can see that the DMD resolution has a slight increase in 1450–2220 nm for the new Hadamard when the order is 107 and 227. However, the resolution increase is not useful because the DMD resolution should match with the spectral resolution. This means if the spectral resolution is lower than the DMD resolution, the resolution of the spectrometer will depend on spectral resolution. In 1300–1450 and 2220–2450 nm, the DMD resolution of the new Hadamard is decreased. By combining SNR with the DMD resolution, we conclude the SNR have been improved by means of sacrificing the resolution in the two sideband. 

## 5. Conclusions

In HT spectrometers with a DMD, we analyze the influence of stray light. It is shown that the stray light mainly affects the spectral energy distribution of the light source. It can reduce the measurement value of the spectral absorbance and make the SNR inconsistent. We address these problems from two aspects. One is to mitigate the impact of the stray light by the double filter strategy, and the other is to improve the SNR by a new Hadamard mask. In addition, the experiments and simulations are conducted and compared between the new scanning method and traditional HT scanning method. The SNRs of the spectrometers are compared for different methods when the order is 107 and 227. The results demonstrate that the new method and mask can effectively suppress the stray light and make the distribution of the spectral energy more uniform. Meanwhile, the SNR also is improved. 

## Figures and Tables

**Figure 1 micromachines-10-00149-f001:**
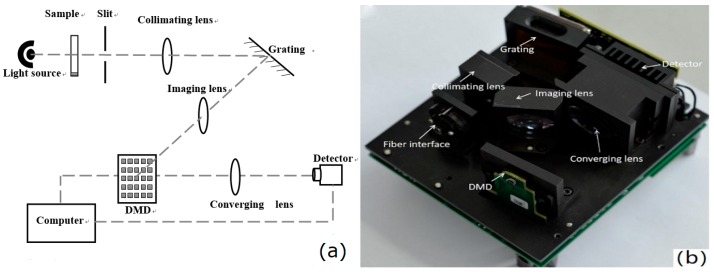
(**a**) Optical system of Hadamard transform (HT) spectrometer. (**b**) Optical structure of HT spectrometer.

**Figure 2 micromachines-10-00149-f002:**
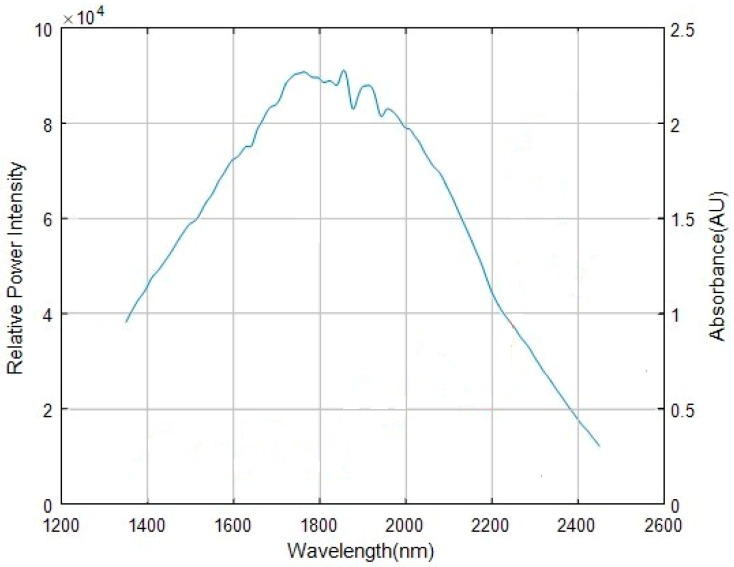
Light source spectrum.

**Figure 3 micromachines-10-00149-f003:**
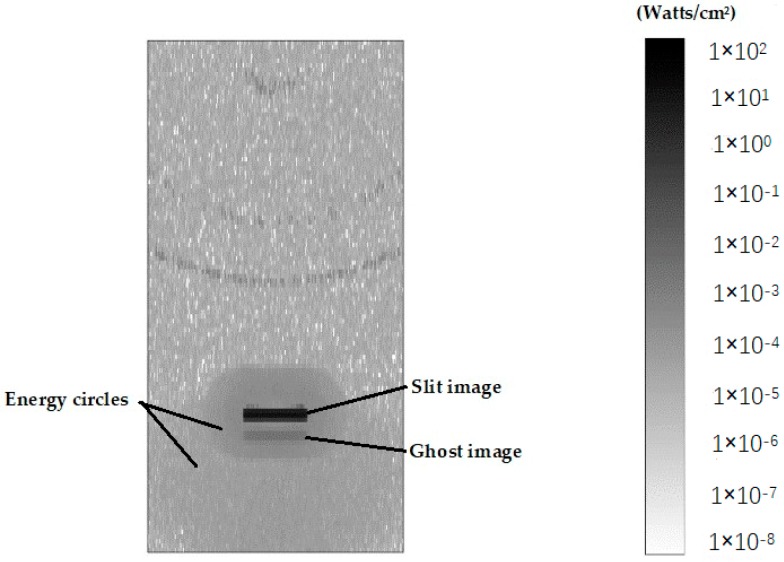
Incoherent radiation of slit image.

**Figure 4 micromachines-10-00149-f004:**
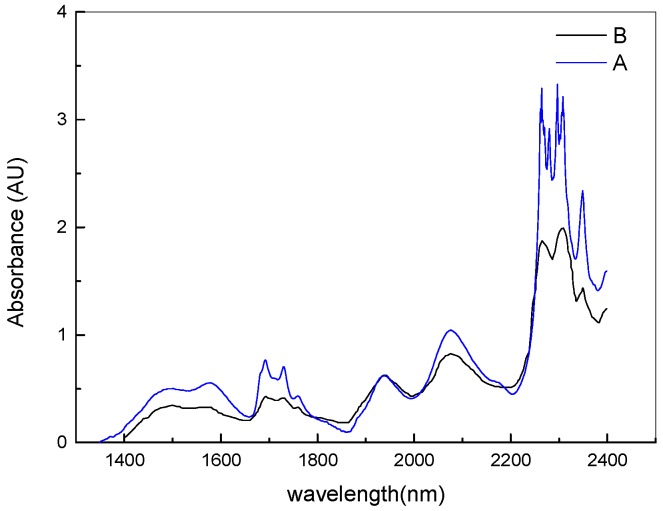
Absorbance spectrum of 95% ethanol. The blue line A is the reference spectrum tested by the UV-visible/NIR spectrophotometer UH4150. The black line B is the absorbance curve of the tested by the traditional scan method.

**Figure 5 micromachines-10-00149-f005:**
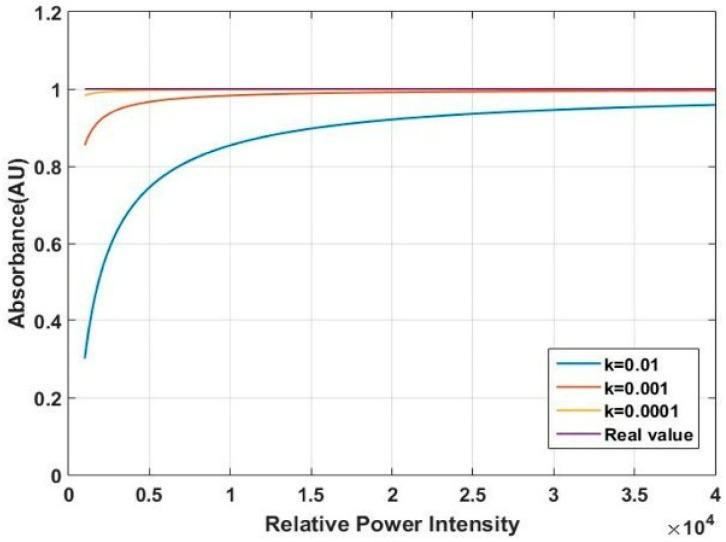
Influence of stray light on the low-energy wavelength band.

**Figure 6 micromachines-10-00149-f006:**
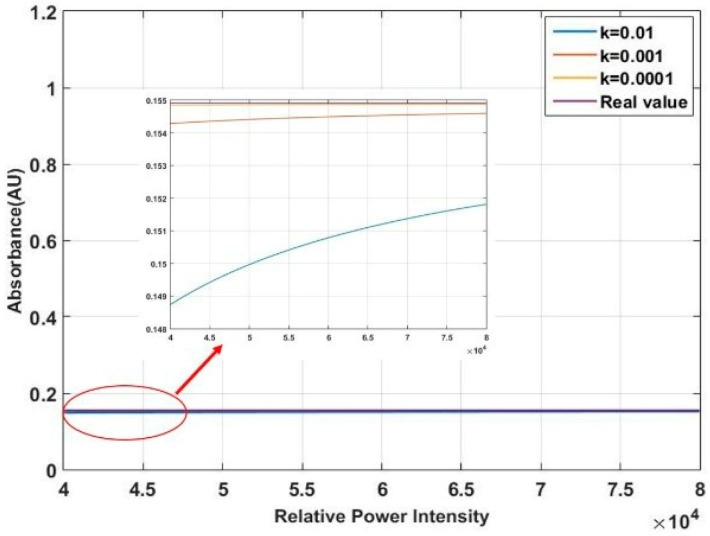
Influence of stray light on the high-energy wavelength Band.

**Figure 7 micromachines-10-00149-f007:**
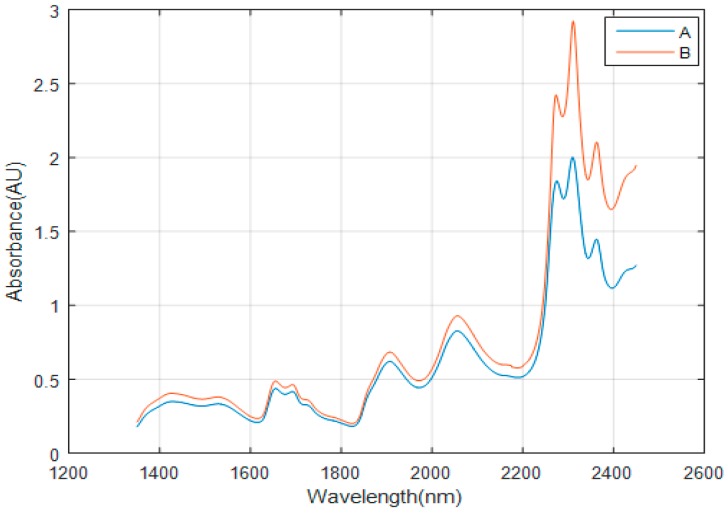
Absorbance spectrum of 95% ethanol. The blue line A is the absorbance curve of the test by the traditional scan method. The red line B is the absorbance curve of the test by the new scan method.

**Figure 8 micromachines-10-00149-f008:**
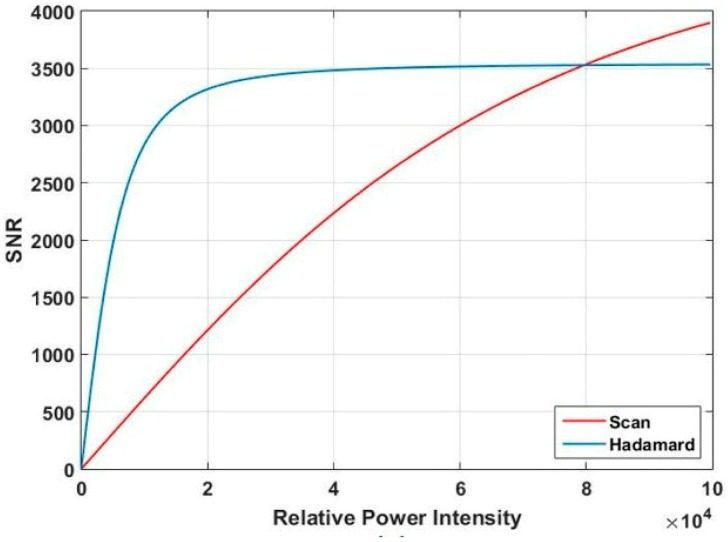
Signal-to-noise ratio (SNR) of two scan methods for light source energy dropping off in proportion.

**Figure 9 micromachines-10-00149-f009:**
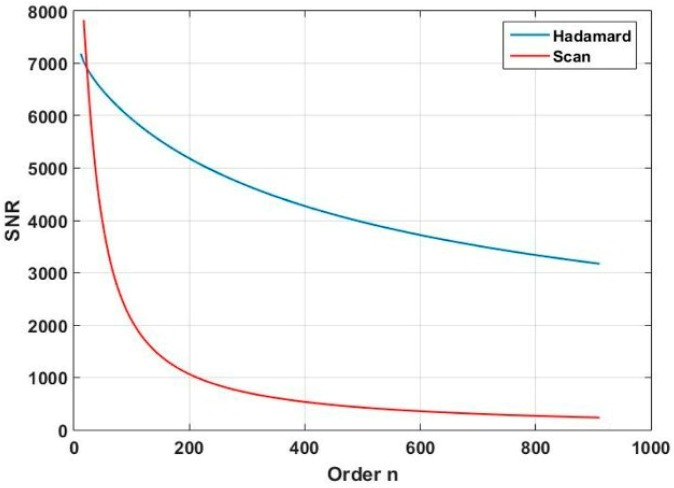
SNR of two scan methods for stable light source.

**Figure 10 micromachines-10-00149-f010:**
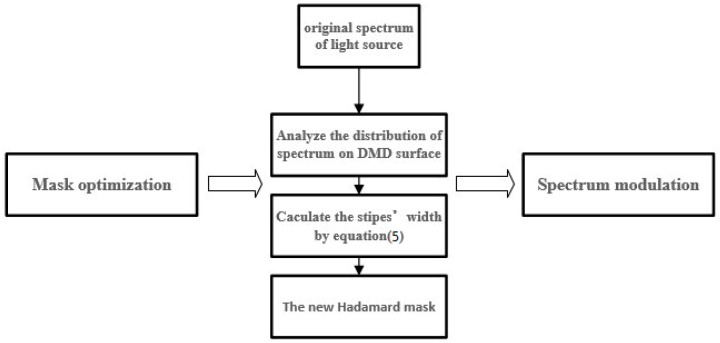
Flowchart of mask optimization.

**Figure 11 micromachines-10-00149-f011:**
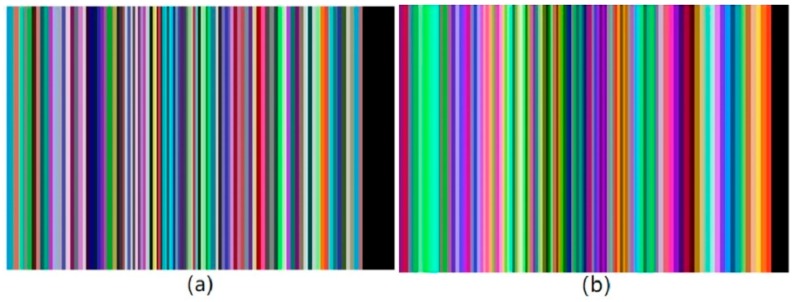
Hadamard masks: (**a**) traditional Hadamard mask (**b**) new Hadamard mask. They are produced by the same order of 107.

**Figure 12 micromachines-10-00149-f012:**
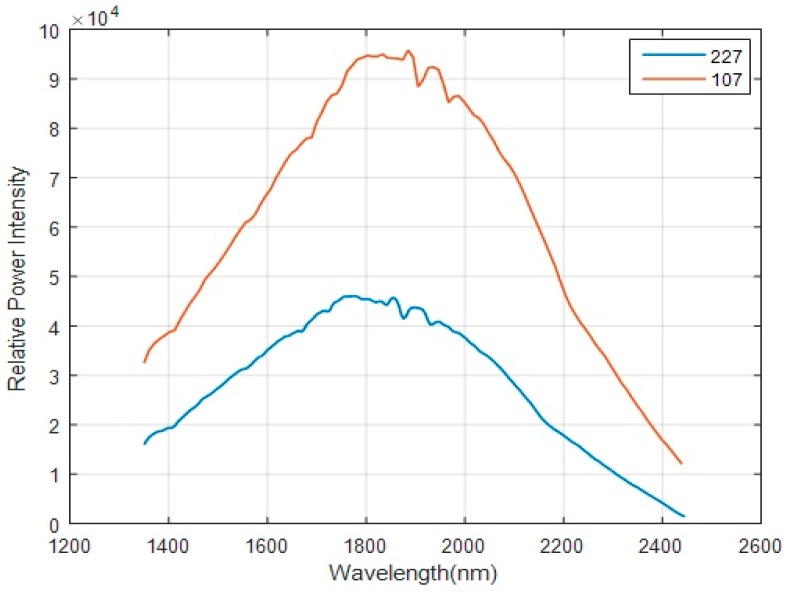
Relative power intensity of the light source. The curves are obtained by the column scan mode with the orders of 107 and 227.

**Figure 13 micromachines-10-00149-f013:**
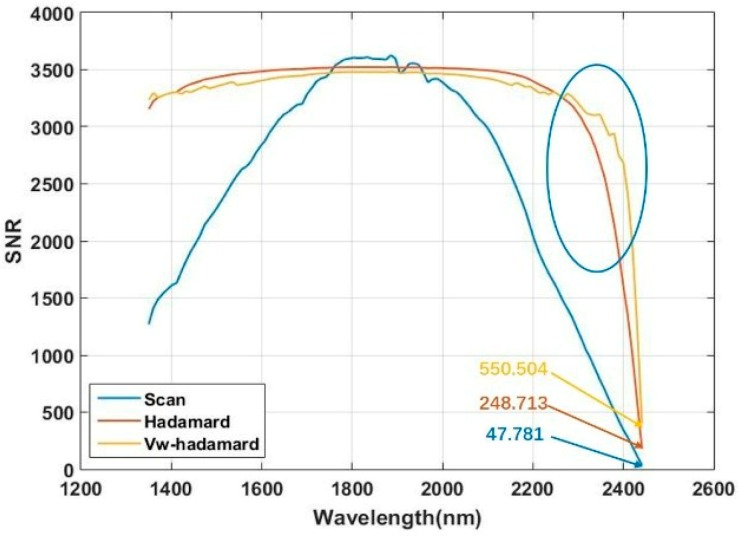
SNR of the spectrometer with the order of 107.

**Figure 14 micromachines-10-00149-f014:**
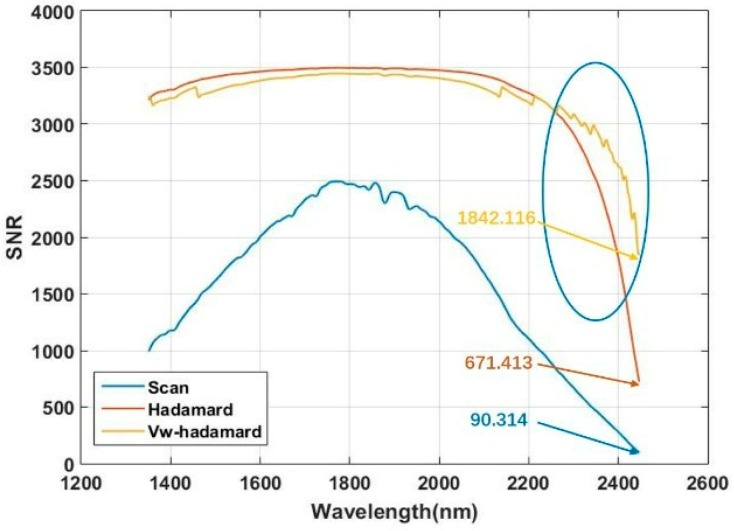
SNR of the spectrometer with the order of 227.

**Figure 15 micromachines-10-00149-f015:**
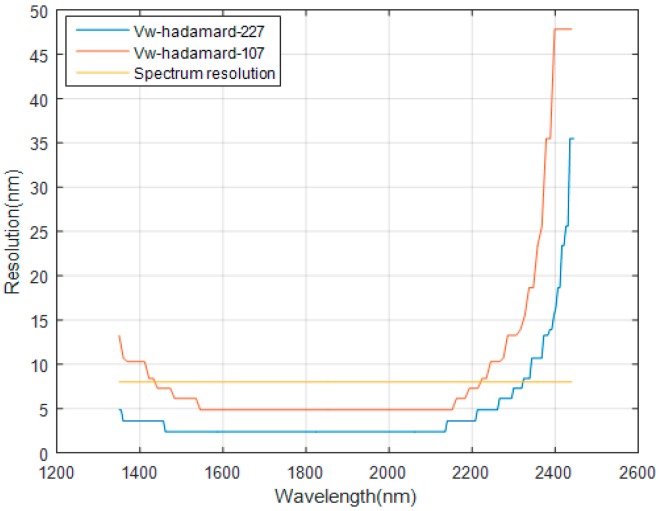
Digital micromirror device (DMD) resolution of the new Hadamard and the traditional Hadamard. The blue and red curves are obtained by the V_W_-Hadamard mask. The yellow curve is the spectral resolution which is the same as the DMD resolution of the traditional Hadamard.

**Table 1 micromachines-10-00149-t001:** Parameters of the HT near-infrared (NIR) spectrometer.

Components	Model/Parameters	Technical Indexes	Parameters
Light source	Tungsten lamp	Spectral range	1350–2450 nm
Slit	50 µm	Spectral resolution	≤ 8 nm
Grating	200 lines/mm	DMD resolution	912 × 1140
DMD	DLP4500NIR	Sampling rate	1 s
DMD mirrors	Aluminum micromirror	-	-
Optical fibers	Infrared quartz	-	-
Detector	InGaAs *R* = 3 mm	-	-

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
