# Peer review of "The Improvement on the Performance of DMD Hadamard Transform Near-Infrared Spectrometer by Double Filter Strategy and a New Hadamard Mask"

_micromachines, 2019, doi:10.3390/mi10020149_

Reviewer 1 Report

Please, find attached the comments.

Author Response

Thank you for your valuable comments, please find our responses in attachment.

Reviewer 2 Report

In this manuscript, researchers have introduced a new analysis method for the spectrometer. The technical writing is well prepared. The literature searching is appropriate. "Minor revision" is suggested before being accepted.

Author Response

Thank you.

Reviewer 3 Report

The manuscript is mainly divided into two parts: the first part is on stray light analysis and how to mitigate its impact, and the second part is on a new Hadamard mask to improve SNR in the long wavelength range. However, the authors didn't make much effort to connect the two parts into one single manuscript, and it seems the two parts should be in two separate papers instead. Moreover, based on the results presented, the improvements from the new methods are not very obvious. Therefore, the reviewer recommend major revision before considering for publication. 

Detailed comments: 

Fig 2. shows the slit image on the detector plane. Is it simulated by Zemax, as the authors mentioned in line 114? If so, could the authors compare with the actual slit image to validate the model? 

Line 119, could the authors elaborate on where does the secondary reflection come from?

Line 121, 'ghosting can be observed under the slit image, which is formed by the window of the DMD'. What is the material of the window, and could anti-reflection coating reduce the ghosting effect?

Line 132, 'It indicates that the energy is higher and the absorbance is lower'. It's not clear what are these metrics compared to. In Fig 3, it will be more useful if the authors could show a reference plot for both the light source spectrum and absorption spectrum of ethanol. 

Line 189, 'As the requirements, our team make two types of filter. The first filter allows high-relative-power waveband (1350–2200 nm) to pass. The other one allows low-relative-power waveband (2200–2450 nm) to pass', Could the authors provide more details, such as how are these filters fabricated, and what are the transmission spectrums of the two filters?

Line 193, 'At the high-relative-power waveband, the curves almost overlap completely.' There is still a noticeable difference in the high power band. Could the authors explain what is causing the difference?

Line 195, 'This indicates that the method of the split waveband scan can suppress parts of the impact of the stray light and make the absorbance increase.', making the absorbance increase doesn't mean the result is more accurate. Could the authors provide a reference spectrum, to show the improvement of the new method?

Fig. 10, could the authors elaborate on how is SNR measured/calculated?

Fig. 11, could the authors compare Vw-Hadamard with the original Hadamard to show the impact to spectral resolution? 

Line 309, 'Meanwhile, the SNR and resolution also is improved'. However, the result does't show that resolution is improved with Vw-Hadamard. 

Extensive language editing required. Examples: 

Line 24, 'the SNR of the spectrometer also increase', should be' the SNR of the spectrometer is also increased'

Table 1, the column with column name 'Model number' doesn't just contain model numbers. Consider using a more accurate column name. 

Line 202, 'NIR analysis technic' should be 'NIR analysis technique'. 

Line 228, 'We must think out' should be 'We must think of'

Author Response

Thank you for your valuable comments, please find our responses in attachment.

Round  2

Reviewer 1 Report

The authors answer my questions.

Author Response

Dear professor, Thank you very much for your comments on our script (Micromachines-4091942). All the comments are valuable and very helpful for revising and improving our paper.

Reviewer 3 Report

The authors have made significant improvements to the manuscript. Below are some additional comments: 

1) In the updated manuscript, figure 4 shows both reference and measured spectrum of 95% ethanol. It seems that both the peak locations and peak intensities are off. The authors explained that the peak intensities are off due to mismatch. Could the authors explain why didn't the peak location match in some cases? For example, reference spectrum has a peak at around 2300nm, but the measured spectrum has a peak at around 2360nm.

2) Could the authors provide some details on how was the spectrometer calibrated, in terms of spectral location and relative irradiance?

Author Response

Dear professor, Thank you very much for your comments on our script (Micromachines-409194). All the comments are valuable and very helpful for revising and improving our paper. Please see our revisions in attachment.
